# Identification of 'erasers' for lysine crotonylated histone marks using a chemical proteomics approach

Xiucong Bao[1†], Yi Wang[2†], Xin Li[1†], Xiao-Meng Li[1†], Zheng Liu[1], Tangpo Yang[1], Chi Fat Wong[3], Jiangwen Zhang[3], Quan Hao[2*], Xiang David Li[1*]

[1]Department of Chemistry, University of Hong Kong, Hong Kong, Hong Kong; [2]Department of Physiology, University of Hong Kong, Hong Kong, Hong Kong; [3]School of Biological Sciences, University of Hong Kong, Hong Kong, Hong Kong

**Abstract** Posttranslational modifications (PTMs) play a crucial role in a wide range of biological processes. Lysine crotonylation (Kcr) is a newly discovered histone PTM that is enriched at active gene promoters and potential enhancers in mammalian cell genomes. However, the cellular enzymes that regulate the addition and removal of Kcr are unknown, which has hindered further investigation of its cellular functions. Here we used a chemical proteomics approach to comprehensively profile 'eraser' enzymes that recognize a lysine-4 crotonylated histone H3 (H3K4Cr) mark. We found that Sirt1, Sirt2, and Sirt3 can catalyze the hydrolysis of lysine crotonylated histone peptides and proteins. More importantly, Sirt3 functions as a decrotonylase to regulate histone Kcr dynamics and gene transcription in living cells. This discovery not only opens opportunities for examining the physiological significance of histone Kcr, but also helps to unravel the unknown cellular mechanisms controlled by Sirt3, that have previously been considered solely as a deacetylase.

**\*For correspondence:** qhao@hku.hk (QH); xiangli@hku.hk (XDL)

[†]These authors contributed equally to this work

**Competing interests:** The authors declare that no competing interests exist.

**Reviewing editor**: Wilfred van der Donk, University of Illinois-Urbana Champaign, United States

## Introduction

Histone posttranslational modifications (PTMs) play a crucial role in regulating a wide range of biological processes, such as gene transcription, DNA replication, and chromosome segregation (*Kouzarides, 2007*). Increasing evidence has indicated that PTMs of histones can serve as a heritable 'code' (so-called 'histone code'), which provides epigenetic information that a mother cell can pass to its daughters (*Jenuwein and Allis, 2001*). Histone code is 'written' or 'erased' by enzymes that add or remove the modifications of histones (*Goldberg et al., 2007*; *Kouzarides, 2007*). Meanwhile, 'readers' of histone code recognize specific histone modifications and 'translate' the code by executing distinct cellular programs necessary to establish diverse cell phenotypes, while the genetic code (DNA) is unaltered (*Seet et al., 2006*; *Taverna et al., 2007*).

Lysine acetylation (Kac) was among the first covalent modification of histones to be described (*Allfrey and Mirsky, 1964*; *Allfrey et al., 1964*). Since its identification, histone Kac has been correlated with gene expression. However, the mechanistic insights into the regulation and functions of histone Kac remained challenging and elusive, until the identification and characterization of the enzymes responsible for the addition and removal of this PTM, which are now known as histone acetyltransferases (*Roth et al., 2001*) and deacetylases (*Sauve et al., 2006*; *Yang and Seto, 2008b*; *Haberland et al., 2009*), respectively. Extensive studies have now revealed that Kac plays an important role in controlling chromatin structure and gene transcription (*Grunstein, 1997*; *Yang and Seto, 2008a*). By neutralizing positively charged lysine residues, acetylation alters the coulumbic interactions between basic histones and the negatively charged DNA, and thereby influences the structure of chromatin compaction (*Ura et al., 1997*; *Shogren-Knaak et al., 2006*). In addition, acetylation may serve as a

**eLife digest** Most of the DNA in a cell is wound around histone proteins to form a compacted structure called chromatin. Enzymes can modify the histones by adding small chemical tags on to them, and these histone modifications can cause the chromatin to either become more tightly packed or more open. Opening up the chromatin makes the DNA more accessible to the cellular machinery involved in gene expression. Thus, cells can regulate which genes they express, and by how much, by modifying the histone proteins.

Like all other proteins, histones are made of smaller molecules called amino acids. Specific amino acids within histone proteins can be modified in a number of different ways, with different effects. For instance, adding a chemical tag called an acetyl group onto an amino acid in a histone weakens the interaction between the histone and the DNA, which opens up the chromatin and increases gene expression.

Another way that histones can be modified is by the addition of crotonyl groups. These chemical tags have not been examined much because the enzymes that add or remove them remain to be identified. However, it was recently suggested that enzymes called sirtuins—which are known to remove acetyl groups from histones—might also remove the crotonyl groups.

Finding histone-modifying enzymes is challenging because the interactions between these enzymes and the histones are both weak and brief. Bao, Wang, Li, Li et al. have now overcome this challenge by developing a method to firmly link any protein that interacts with a crotonylated histone to the histone. Three out of the seven sirtuin enzymes found in humans were revealed to bind to crotonylated histones. All three of these enzymes—called Sirt1, Sirt2 and Sirt3—could remove crotonyl groups from histones in a test-tube, and Sirt3 could also do the same in living cells. Further biochemical experiments suggested that the mechanism used by these enzymes to remove crotonyl groups is the same as the mechanism they use to remove acetyl groups.

Bao, Wang, Li, Li et al. then uncovered the three-dimensional structure of the Sirt3 enzyme bound to a crotonylated histone, and revealed that the enzyme recognizes the crotonyl group on the histone via a unique interaction between the crotonyl group and a specific amino acid in the binding pocket of Sirt3. This amino acid is also found in Sirt1 and Sirt2, but not in other sirtuins; this interaction can thus explain why decrotonylation activity was only detected for these three enzymes.

Moreover, the levels of crotonylated histones and gene expression were higher in cells that lacked Sirt3, but not in those lacking Sirt1 or Sirt2. By identifying Sirt3 as the main decrotonylation enzyme in living cells, the role of histone crotonylation can now be investigated in greater detail.

docking site for 'reader' proteins (e.g., bromodomain containing proteins), which are recruited onto chromatin to carry out downstream cellular processes, such as gene transcription (*Dhalluin et al., 1999*; *Marmorstein and Berger, 2001*; *Zeng et al., 2010*).

Lysine crotonylation (Kcr) is a newly discovered histone PTM that is specifically enriched at active gene promoters and potential enhancers in mammalian cell genomes (*Tan et al., 2011*). In postmeiotic male germ cells, Kcr specifically marks testis specific X-linked genes, suggesting it is likely that it is an important histone mark for male germ cell differentiation. However, further mechanistic and functional studies of histone Kcr have been limited by a lack of knowledge of the enzymes that catalyze the addition or removal of Kcr in cells. In a systematic screening of the activities of the 11 human zinc-dependent lysine deacetylases (i.e., HDAC1–HDAC11) against a series of C-terminal lysine acylated peptides, Olsen et al. found that HDAC3 in complex with nuclear receptor corepressor 1 (HDAC3–NCoR1) had detectable decrotonylase activity towards a model peptide substrate in a fluorometric assay (*Madsen and Olsen, 2012*). Recently, using a radioactive thin layer chromatography based assay, Denu et al. demonstrated that Sirt1 and Sirt2 can catalyze the removal of a crotonyl group from a histone H3K9Cr peptide (*Feldman et al., 2013*). However, this discovery was based on a single peptide substrate. Due to lack of further characterization of these identified enzymes, their mechanisms of catalysis and the molecular bases of substrate recognition remain unclear. More importantly, since both discoveries relied on peptide based in vitro screening assays, there is still an essential need to identify endogenous histone decrotonylases.

To fill this knowledge gap, a method to profile 'eraser' enzymes that recognize Kcr is needed. A Cross-Linking Assisted and Stable isotope labeling of amino acids in cell culture (SILAC) based Protein Identification (CLASPI) approach has recently been reported to identify histone PTM 'readers' (*Li et al., 2012*; *Li and Kapoor, 2010*). However, this approach has not previously been explored to identify histone PTM 'erasers', which are likely involved in weak and transient interactions. Here we present the application of an optimized CLASPI approach to comprehensively profile 'eraser' enzymes that recognize histone Kcr marks. We identified human Sirt1, Sirt2, and Sirt3 as decrotonylases in vitro and examined the molecular basis for how the enzymes recognize Kcr using X-ray crystallography. Furthermore, we demonstrated that Sirt3 can function as an 'eraser' enzyme to regulate histone crotonylation dynamics in living cells.

## Results

### Chemical proteomics approach to profile proteins recognizing histone H3K4Cr mark

We first focused on a crotonylation mark discovered on histone H3K4 (*Tan et al., 2011*). We designed a peptide probe (probe **1**, *Figure 1A*) to convert non-covalent protein–protein interactions mediated by this Kcr into irreversible covalent linkages through photo-cross-linking. The probe is based on the unstructured N-terminal region of histone H3, with lysine-4 crotonylated, a photo-cross-linker (benzophenone) appended to alanine-7, and a bio-orthogonal handle (alkyne) at the peptide C terminus to enable selective isolation of captured binding partners. To identify proteins that bind H3K4Cr with high selectivity and high affinity, we performed two types of CLASPI experiments with cell lysates derived from HeLa S3 cells grown in medium containing either 'heavy' ($^{13}$C, $^{15}$N-substitued arginine and lysine) or 'light' (natural isotope abundance forms) amino acids (*Figure 1B*).

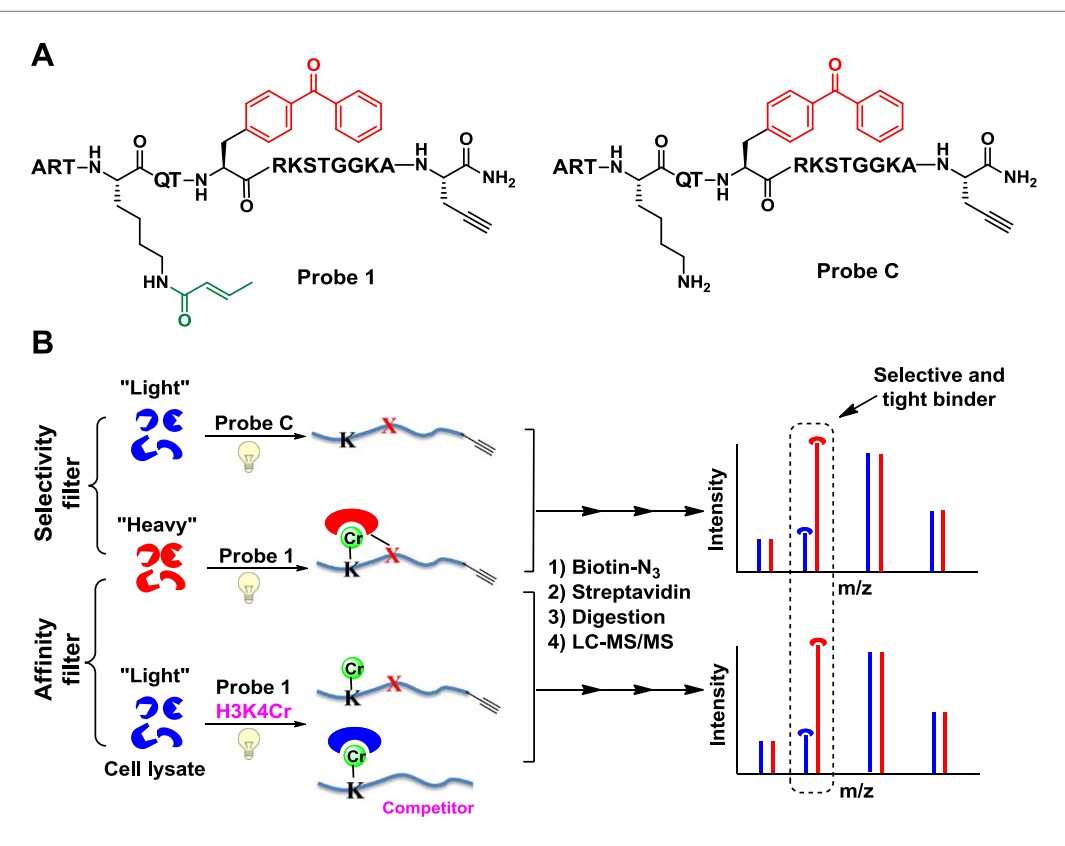

**Figure 1**. Cross-Linking Assisted and SILAC based Protein Identification (CLASPI) strategy. (**A**) Chemical structures of probe **1** and probe **C**. (**B**) Schematic diagram illustrating the CLASPI strategy to profile proteins that bind H3K4Cr with high selectivity and affinity in whole-cell proteomes. LC-MS, liquid chromatography–mass spectrometry.

In a 'selectivity filter' CLASPI experiment, the 'heavy' and 'light' cell lysates were photo-cross-linked with probe **1** and an unmodified H3 control probe (probe **C**, *Figure 1A*), respectively, and pooled for the subsequent steps. The captured proteins were then conjugated to biotin using click chemistry, followed by affinity purification, gel electrophoresis, and in-gel trypsin digestion. The digested peptide mixtures were separated by high performance liquid chromatography (HPLC) and analyzed with a LTQ-Orbitrap mass spectrometer. Using this method, proteins that show a high SILAC ratio of heavy/light (H/L) are likely H3K4Cr selective binders. To further distinguish the high affinity interactions, we performed an 'affinity filter' CLASPI experiment, in which both lysates were photo-cross-linked with probe **1** but the 'light' sample also contained H3K4Cr peptide as a competitor (30 μM) (*Figure 1B*). We expected that the addition of the competitor peptide in the 'light' lysate would effectively inhibit **1**-induced cross-linking of H3K4Cr binders that have high affinity ($K_d < 30$ μM) towards the H3K4Cr peptide, and should thereby produce a high SILAC ratio of H/L for these proteins. Together, we consider a protein as a selective and tight binder of H3K4Cr when it shows high SILAC ratios of H/L in both 'selectivity filter' and 'affinity filter' experiments (*Figure 2—source data 1*).

## Sirt1, Sirt2, and Sirt3 recognize histone H3K4Cr mark

A two-dimensional plot with logarithmic (Log$_2$) SILAC ratios of H/L of the identified proteins in the 'selectivity filter' and 'affinity filter' experiments, along the x axis and y axis, respectively, is shown in *Figure 2A*. As expected, the majority of identified proteins did not show significant differences between the signal intensities of their 'heavy' and 'light' forms (i.e., H/L close to 1:1), suggesting they are not likely to be H3K4Cr binding proteins. In contrast, three nicotinamide adenine dinucleotide (NAD)-dependent deacetylases (*Imai et al., 2000*; *Landry et al., 2000*; *Sauve et al., 2006*), Sirt1, Sirt2, and Sirt3, were enriched by more than 10-fold by the K4 crotonylated probe (**1**) in the 'selectivity filter' experiment (*Figure 2A, B* and *Figure 2—figure supplement 1*), indicating that they preferentially bind to this histone Kcr mark. However, among these three selective H3K4Cr binders, only Sirt3 showed the highest SILAC ratio of H/L and thereby appeared as an outlier outside of the background in the 'affinity filter' experiment (*Figure 2A, B* and *Figure 2—figure supplement 1*). This result indicates that Sirt3 is likely a selective and relatively tight binding partner of H3K4Cr.

We next examined whether Sirt3 can directly and selectively bind to this crotonylated histone peptide in vitro. As shown in *Figure 2C*, the recombinant Sirt3 was captured by probe **1** but not by probe **C**, and the cross-linking was competed by the H3K4Cr peptide with an IC$_{50}$=32.3 μM (*Figure 2D*), verifying a direct and selective interaction between Sirt3 and the K4 crotonylated H3 peptide. Indeed, the direct measurement of binding affinity using isothermal titration calorimetry showed that Sirt3 bound to the H3K4Cr peptide with $K_d$=25.1 μM (*Figure 2E*). Consistent with our 'affinity filter' CLASPI analysis, Sirt1 and 2 showed lower affinities towards the H3K4Cr peptide (*Figure 2—figure supplement 2*), indicating that they are selective but relatively weak binders towards this histone Kcr mark.

## Molecular basis for how Sirt3 recognizes histone Kcr

To study the molecular basis for the recognition of H3K4Cr by Sirt3, we determined the crystal structure of human Sirt3 in complex with an H3K4Cr peptide to 2.95 Å resolution (PDB 4V1C). The asymmetric unit consists of six molecules, each containing one Sirt3–H3K4Cr complex. The two globular domains of Sirt3 composed of an NAD binding Rossmann fold and a zinc binding motif are similar to other sirtuins (*Figure 3A*) (*Avalos et al., 2004*; *Du et al., 2011*; *Yuan and Marmorstein, 2012*; *Jiang et al., 2013*). Residues [2]RTKQTAR[8] of the H3K4Cr peptide were clearly identified based on electron density. The way that the substrate is bound is similar to the published complex structure of Sirt3, with a lysine acetylated AceCS2 peptide (PDB 3GLR) (*Figure 3—figure supplement 1*) (*Jin et al., 2009*). The crotonyl lysine is located in a binding pocket formed by hydrophobic residues Phe180, Ile230, His248, Ile291, and Phe294 of Sirt3 (*Figure 3B*). Residue His248, a catalytic residue for the deacetylation activity of Sirt3, interacts with the crotonyl amide oxygen via hydrogen bonding in the structure (*Figure 3B*). Strikingly, the phenyl ring of residue Phe180 aligns parallel to the planar crotonyl group and has a short distance of 3.6 Å to its conjugated carbon–carbon double bond (C=C) (*Figure 3C, D*), indicating a robust π-π stacking interaction between the two functional groups. Interestingly, a primary sequence alignment of all sirtuins revealed that the phenylalanine residue (Phe180) of Sirt3 is conserved in Sirt1 and Sirt2, but not in other sirtuins (*Figure 3—figure supplement 2*), which may explain why Sirt4–Sirt7 were not identified in our CLASPI experiments. This π-π interaction therefore underlies the mechanism for the recognition of crotonyl lysine by Sirt1, Sirt2, and Sirt3.

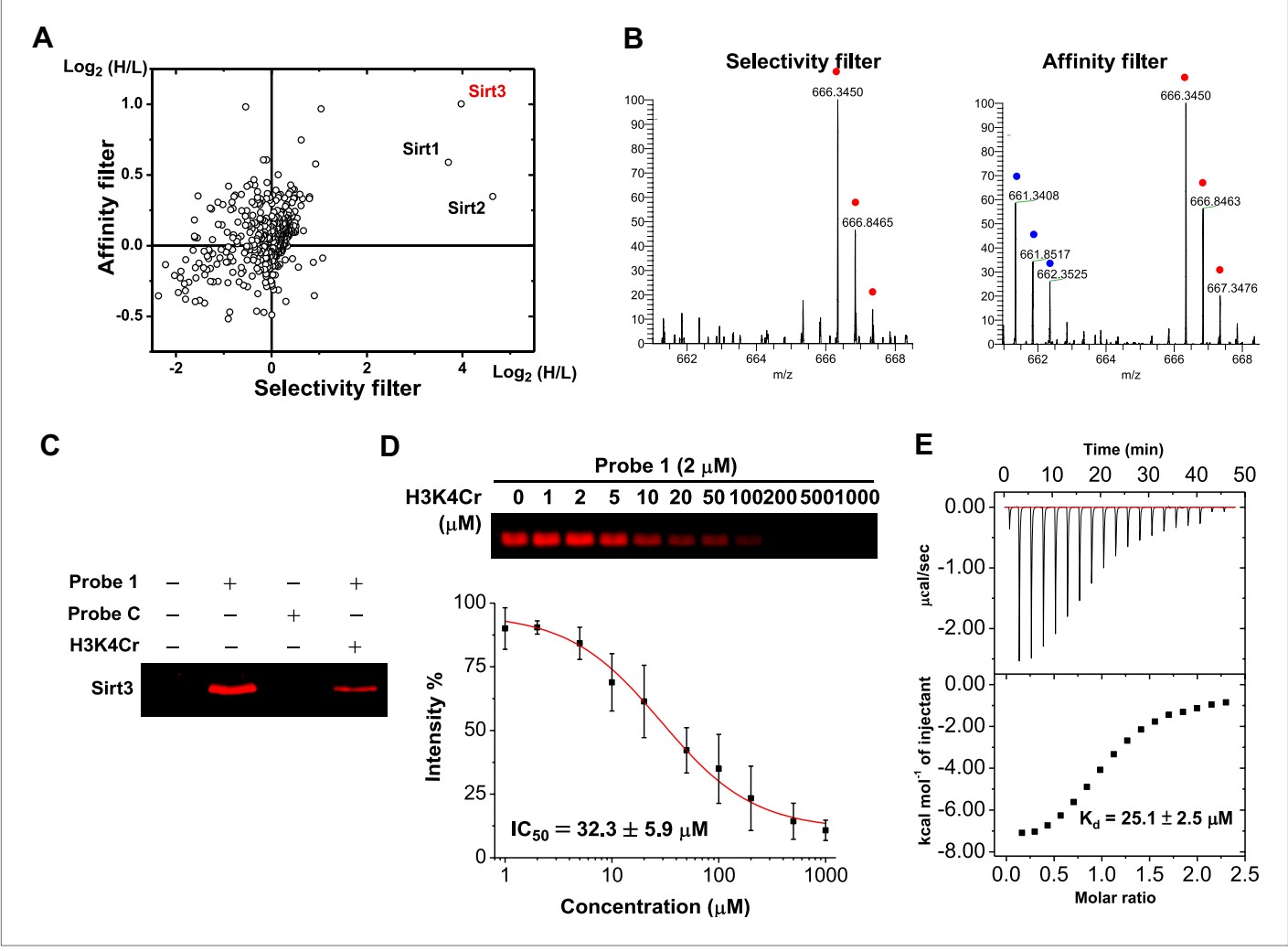

**Figure 2**. Identification of Sirt3 as a selective and tight binding partner of lysine-4 crotonylated histone H3. (**A**) A two-dimensional plot showing the $Log_2$ values of the stable isotope labeling of amino acids in cell culture (SILAC) ratios (heavy/light (H/L)) of each identified protein for the 'selectivity filter' (x axis) and 'affinity filter' (y axis) experiments. (**B**) Representative mass spectrometry (MS) spectra of a peptide, [225]LYTQNIDGLER[235], from Sirt3 identified in both the 'selectivity filter' and 'affinity filter' experiments. The 'light' and 'heavy' peptide isotopes are indicated by blue and red dots, respectively. (**C**) Recombinant Sirt3 was selectively labeled in vitro by crotonylated probe **1** (2 µM) and the labeling by probe **1** was inhibited by a H3K4Cr peptide (30 µM). (**D**) Determination of $IC_{50}$ for inhibition of probe **1** induced labeling of Sirt3 by H3K4Cr peptide (n=3, mean±s.e.). (**E**) Isothermal titration calorimetry measurement for the binding affinity of Sirt3 towards the H3K4Cr peptide.

The following source data and figure supplements are available for figure 2:

**Source data 1**. Proteins quantified in the 'selectivity filter' and 'affinity filter' Cross Linking Assisted and SILAC based Protein Identification (CLASPI) experiments.

**Figure supplement 1**. Representative mass spectrometry spectra for peptides from Sirt1 and Sirt2.

**Figure supplement 2**. ITC measurement for the binding affinity of Sirt1-3 toward H3K4Cr peptide.

## Sirt1, Sirt2, and Sirt3 catalyze hydrolysis of crotonylated histone peptides in vitro

Inspired by the fact that Sirt3 binds crotonyl lysine at its catalytic pocket that is known for hydrolysis of acetyl lysine, we next tested whether Sirt3 has decrotonylation activity. Liquid chromatography–mass spectrometry (LC-MS) was used to monitor hydrolysis of the H3K4Cr peptide by Sirt3. As

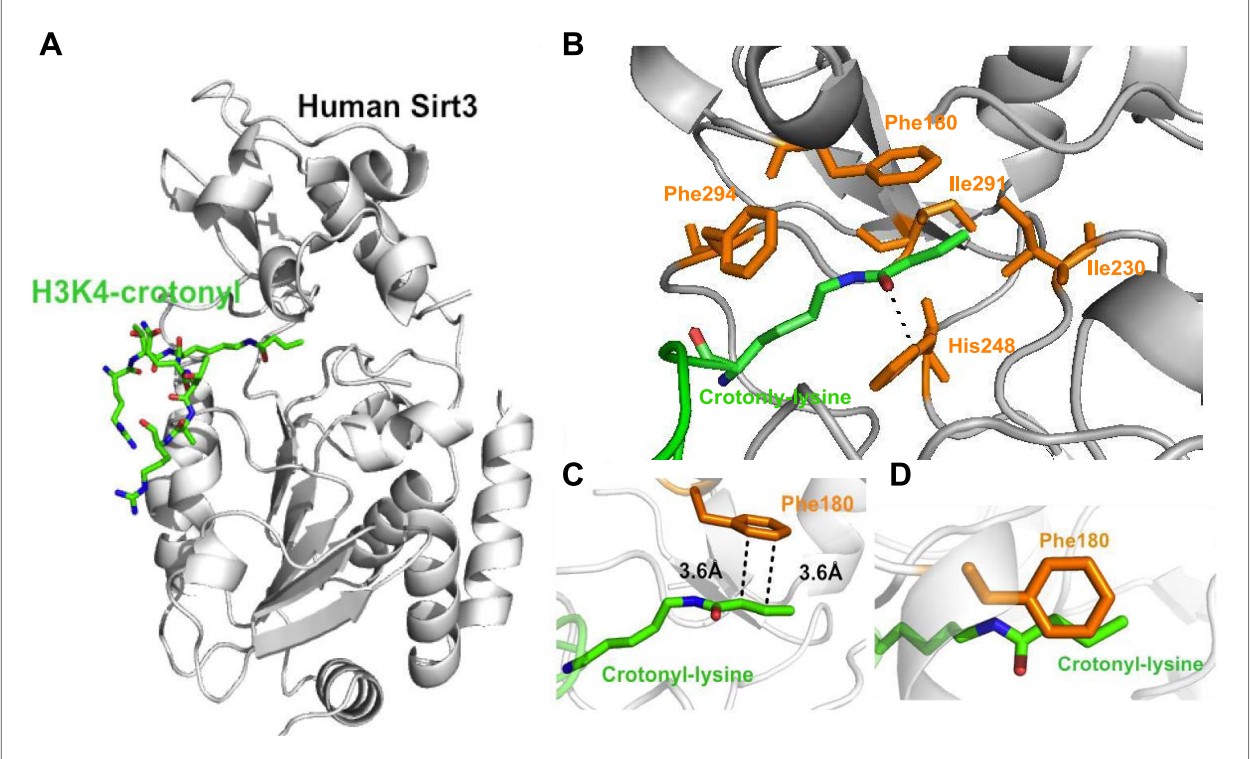

**Figure 3**. Structural basis for how Sirt3 recognizes lysine crotonylation. (**A**) Overall structure of the complex of Sirt3 (gray) with H3K4Cr peptide (green). (**B**) The binding pocket formed by hydrophobic residues (orange) that accommodate the crotonyl lysine. A side view (**C**) and top view (**D**) of a π-π stacking interaction between residue Phe180 of Sirt3 and the crotonyl group.

The following figure supplements are available for figure 3:

**Figure supplement 1**. Detailed structural analysis for Sirt3 in complex with H3K4Cr peptide.

**Figure supplement 2**. Sequence alignment of human Sirt1–Sirt7.

expected, Sirt3 efficiently catalyzed the hydrolysis of the crotonyl peptide only in the presence of NAD (*Figure 4A*), suggesting an NAD-dependent decrotonylation mechanism. The steady state kinetic analysis revealed that the $k_{cat}$, $K_m$, and $k_{cat}/K_m$ for Sirt3 catalyzed decrotonylation of H3K4Cr were 0.010 s$^{-1}$, 12.6 μM, and 783 s$^{-1}$ M$^{-1}$, respectively (*Figure 4—figure supplement 1*). In addition, we detected *O*-crotonyl-adenosine 5'-diphosphoribose (*O*-Cr-ADPR) as a product of this hydrolysis reaction (*Figure 4—figure supplement 2*). A mutation of the catalytic residue (H248Y) that is crucial for the deacetylation activity of the enzyme also completely abolished its decrotonylation activity (*Figure 4C*). These data indicate that Sirt3 hydrolyzes crotonyl lysine with the same mechanism as it hydrolyzes acetyl lysine (*Figure 4—figure supplement 3*) (*Tanner et al., 2000*; *Tanny and Moazed, 2001*). In addition to H3K4Cr, we also examined the activity of Sirt3 to hydrolyze a collection of crotonyl histone peptides (*Tan et al., 2011*). As shown in *Figure 4D–G*, Sirt3 manifested varied decrotonylation activities towards these peptides and this substrate selectivity can be partially explained by the binding affinities of Sirt3 to these peptides (*Figure 4—figure supplement 4*). The observation that Sirt3 binds a crotonylated peptide by recognizing both the modification site and its surrounding residues was also supported by the extensive hydrophobic and hydrogen bonding interactions between Sirt3 and the peptide side chains in the Sirt3–H3K4Cr complex structure (*Figure 3—figure supplement 1B*).

We next investigated whether other members of the sirtuin family could also function as decrotonylases. Consistent with the work of Denu and coworkers, Sirt1 and Sirt2 also catalyzed the hydrolysis of the H3K4Cr peptide, although they were relatively weaker binders towards this substrate (*Figure 2—figure supplement 2*). In contrast, for Sirt5 or Sirt6, little hydrolysis of the crotonyl peptide was observed (*Figure 4—figure supplement 5*). These results agree well with the observation that the

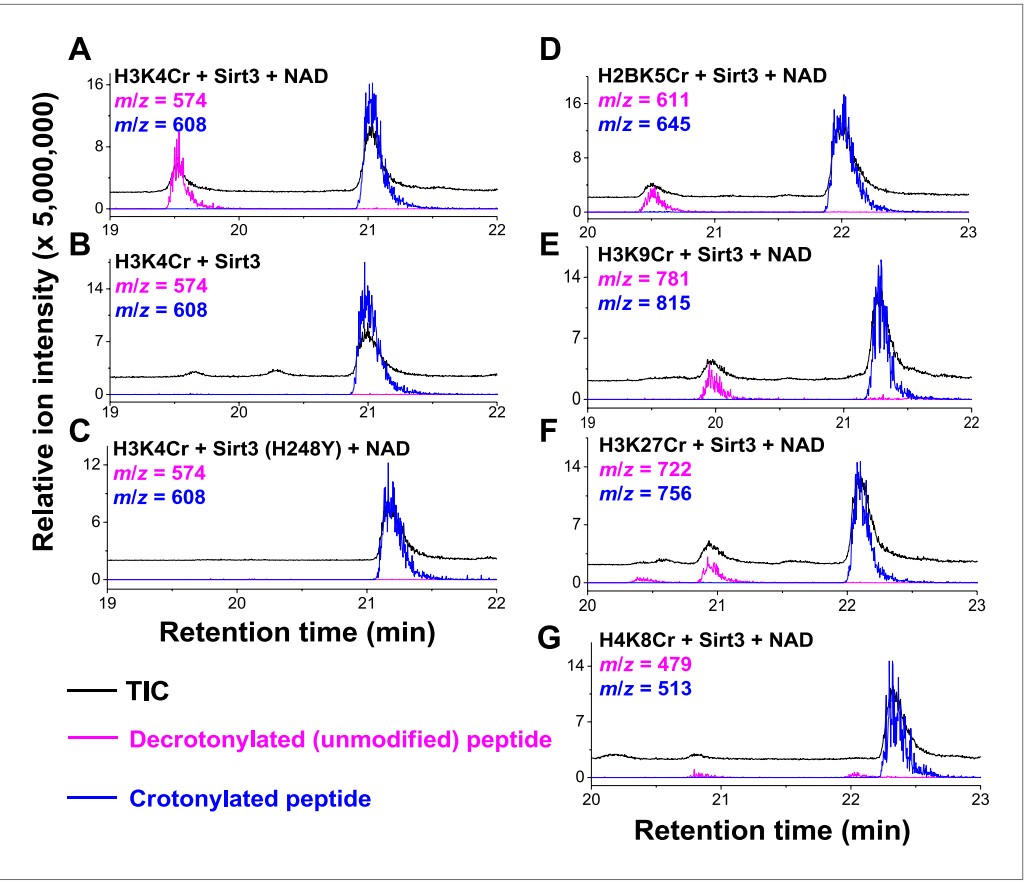

**Figure 4**. Sirt3 catalyzes the hydrolysis of crotonyl lysine in vitro. (**A**–**C**) The hydrolysis of the crotonylated peptides by Sirt3 was analyzed by liquid chromatography–mass spectrometry. The hydrolysis of H3K4Cr was observed with Sirt3 in the presence (**A**), but not absence of nicotinamide adenine dinucleotide (NAD) (**B**), or with the mutated Sirt3, H248Y (**C**). (**D**–**G**) Sirt3 showed varied decrotonylation activities towards H2BK5Cr (**D**), H3K9Cr (**E**), H3K27Cr (**F**), and H4K8Cr (**G**) peptides. Black traces show total ion intensity for all ion species with *m/z* from 300 to 2000 (i.e., total ion counts, TIC); pink traces show ion intensity (5× magnified) for the masses of decrotonylated (unmodified) peptides; and blue traces show ion intensity (5× magnified) for the masses of crotonylated peptides.

The following figure supplements are available for figure 4:

**Figure supplement 1**. Michaelis–Menten plots showing the kinetics of Sirt3 and mutant Sirt3 (F180L) decrotonylation on H3K4Cr.

**Figure supplement 2**. Detection of *O*-crotonyl-adenosine 5'-diphosphoribose (*O*-Cr-ADPR) by liquid chromatography–mass spectrometry (LC-MS).

**Figure supplement 3**. Proposed mechanism of Sirt3 catalyzed nicotinamide adenine dinucleotide dependent decrotonylation.

**Figure supplement 4**. ITC measurement for the binding affinity of Sirt3 toward crotonylated histone peptides.

**Figure supplement 5**. Decrotonylation activity of sirtuins in vitro. The hydrolysis of the H3K4Cr peptide by sirtuins was analyzed by liquid chromatography–mass spectrometry.

**Figure supplement 6**. Michaelis–Menten plots showing the kinetics of Sirt3 and mutant Sirt3 (F180L) deacetylation on H3K4Ac.

phenylalanine residue, which is involved in recognition of crotonyl lysine via π–π stacking interaction (*Figure 3C, D*), is only conserved in Sirt1–Sirt3 (*Figure 3—figure supplement 2*). To further examine the importance of this conserved phenylalanine to the decrotonylase activity of the enzyme, we mutated Phe180 of Sirt3 to a leucine residue (F180L), which lacks an aromatic ring as a π donor but retains a similar hydrophobicity. We then carried out kinetic studies on this F180L mutant Sirt3. The steady state kinetic data showed that the catalytic efficiency of Sirt3 F180L mutant ($k_{cat}/K_m$=21 s$^{-1}$ M$^{-1}$) for the hydrolysis of the H3K4Cr peptide was about 40-fold lower than that of wild-type Sirt3 (*Figure 4—figure supplement 1*), indicating a critical role of the phenylalanine mediated π–π interaction in the decrotonylation activity of the enzyme. Interestingly, the F180L mutation caused only about a two-fold decrease in the deacetylation activity of the enzyme (*Figure 4—figure supplement 6*). This result rules out the possibility that the observed significant decrease in the decrotonylase activity of the enzyme is caused by a potential disruption of the NAD binding pocket in the mutated Sirt3.

## Sirt1, Sirt2, and Sirt3 remove Kcr marks from histone proteins in vitro

To test whether Sirt1–Sirt3 decrotonylate proteins, we incubated whole-cell proteins that were resolved in a sodium dodecyl sulfate–polyacrylamide gel electrophoresis (SDS-PAGE) gel and transferred onto a poly(vinylidene fluoride) (PVDF) membrane with the enzymes in the presence of NAD. A pan antibody against Kcr was used to assess protein crotonylation levels. While the incubations with Sirt1–Sirt3 had little influence on lysine crotonylation in most of the protein bands, substantial reductions in Kcr levels were observed in two bands with a molecular mass of approximately 15 kDa (*Figure 5A*). Considering Sirt1–Sirt3 can decrotonylate histone peptides in vitro, we speculated that these 15 kDa proteins with reduced Kcr levels could be histones. We therefore examined the decrotonylation activity of Sirt1–Sirt3 using purified core histone proteins as substrates. Indeed, Sirt1–Sirt3 not only reduced global Kcr levels of all core histones, they also showed robust decrotonylation activity towards two known histone Kcr sites, H3K4Cr and H3K27Cr (*Figure 5B* and *Figure 5—figure supplement 1*).

## Sirt3 regulates histone Kcr levels in cells

We next examined whether Sirt1–Sirt3 regulate histone lysine crotonylation in cells. Although Sirt1 and Sirt2 can decrotonylate histone peptides and proteins in vitro, their knockdowns by siRNA did not cause an appreciable increase in crotonylation levels for both the global histone and the two tested Kcr (i.e., H3K4Cr and H3K27Cr) sites (*Figure 5—figure supplement 2*). In contrast, Sirt3 knockdown caused accumulation of global histone crotonylation and the H3K4Cr mark, while the histone H3K4Ac and H3K4Me3 levels were unaltered (*Figure 5C, D*), suggesting that Sirt3 selectively targets histone crotonylation. Interestingly, the crotonylation level on H3K27 was not influenced by the knockdown of Sirt3, which may be explained by the observation that Sirt3 showed weaker activity towards the H3K27Cr peptide in vitro (*Figure 4—figure supplement 3*). It should be noted that Sirt3 was found to localize predominantly to mitochondria and was mainly involved in metabolic regulations through controlling protein acetylation dynamics. However, recent evidence has suggested that Sirt3 can also be present in the nucleus in its full length form (*Scher et al., 2007*; *Iwahara et al., 2012*). Indeed, using an antibody that targets the N-terminal region of Sirt3, we detected endogenous full length Sirt3 in the nucleus of HeLa cells by both immunofluorescence and western blotting analyses (*Figure 5—figure supplement 3*). Taken together, these data suggest that endogenous Sirt3 can function as an 'eraser' enzyme to regulate histone crotonylation dynamics in cells.

## Sirt3 regulates histone Kcr levels and gene expression on its defined chromatin regions

Finally, we sought to determine the potential biological consequence of histone decrotonylation mediated by Sirt3. It has been reported that Sirt3 can bind to chromatin and cause repression of the neighboring genes in U2OS cells (*Iwahara et al., 2012*). We therefore hypothesized that Sirt3 could regulate gene transcription via controlling local histone Kcr levels. To test this hypothesis, we focused on seven candidate genes, *Baz2a*, *Brip1*, *Corin*, *Ptk2*, *Tshz3*, *Wapal*, and *Zfat*, whose transcription start sites are close to the Sirt3 enriched region. Chromatin precipitation (ChIP) coupled with quantitative PCR (qPCR) was performed in U2OS cells with the pan anti-Kcr antibody to measure Kcr levels near the transcription start sites of the candidate genes. As shown in *Figure 5E*, Sirt3 knockdown by siRNA resulted in significant increases in Kcr levels of five of the seven genes analyzed, indicating that Sirt3 may directly regulate crotonylation dynamics at the genomic loci where it binds. Interestingly, the mRNA levels of the three candidate genes with increased Kcr levels, *Ptk2*, *Tshz3*, and *Wapal*, were also

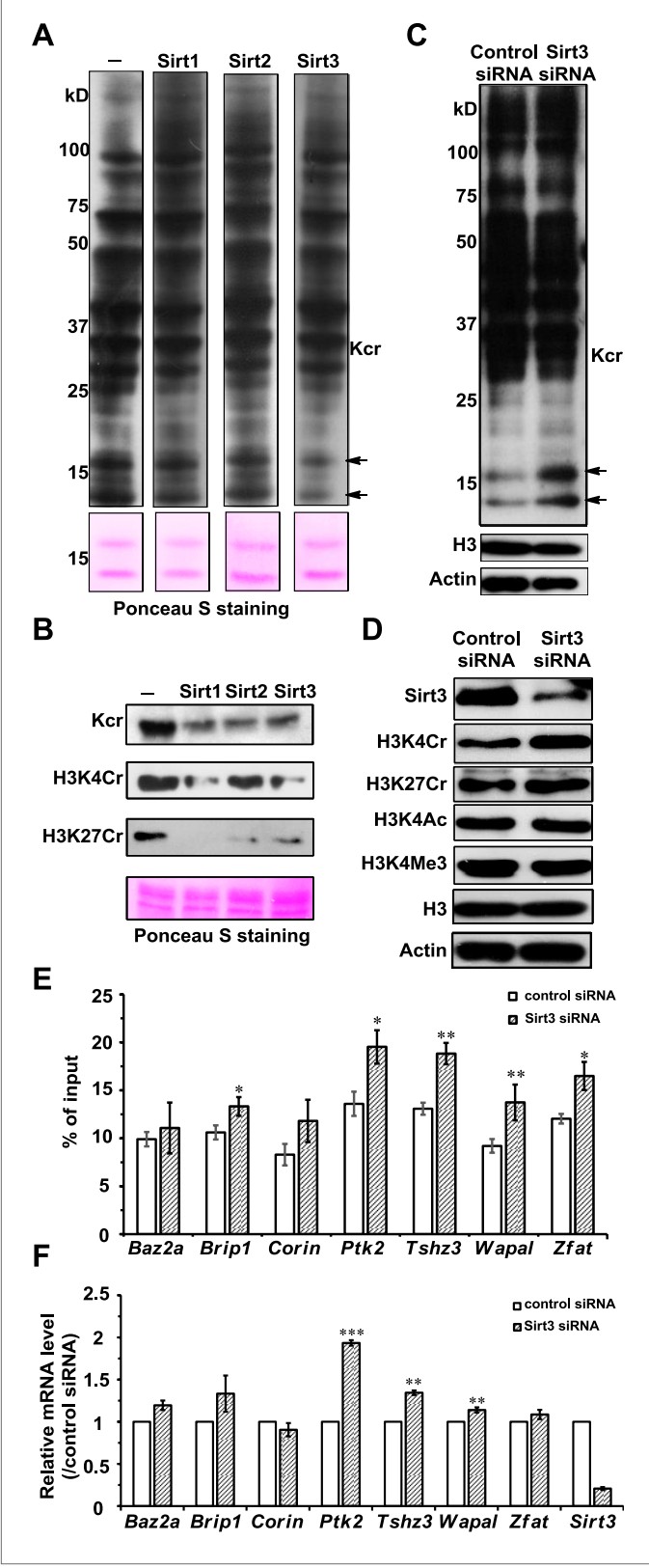

**Figure 5**. Sirt3 regulates histone lysine crotonylation and gene expression. (**A**) Western blot analyses showing Sirt1–3 catalyzed decrotonylation of whole-cell lysates on the membrane. The arrows indicate the protein bands with reduction of crotonylation levels. Ponceau S staining was used as the loading control. (**B**) Western blot
*Figure 5. Continued on next page*

*Figure 5. Continued*

analyses showing the decrotonylation of purified histones catalyzed by Sirt1–3 in the reaction buffer. Ponceau S staining was used as the loading control. (**C**) Western blot analyses showing that Sirt3 knockdown caused the accumulations of H3K4Cr without effect on H3K4Ac, H3K4Me3, or H3K27Cr levels. (**D**) Western blot analyses showing that Sirt3 knockdown caused global histone crotonylation (as indicated by the arrows). H3 and actin were used as loading controls. (**E**) Chromatin immunoprecipitation–quantitative PCR analyses showing the changes in the histone Kcr levels of the indicated chromatin loci on Sirt3 knockdown. Quantitative PCR signal was normalized by a non-Sirt3 bound region of *Gapdh*. (**F**) Real time PCR analyses showing the changes in mRNA level of the indicated genes on Sirt3 knockdown. Quantitative PCR signal was normalized by *Gapdh*. Error bars indicated ± s.e. from four (**E**) or three (**F**) independent biological replicates. The p values are based on the Student's t test. *$p<0.05$, **$p<0.01$, ***$p<0.001$.

The following figure supplements are available for figure 5:

**Figure supplement 1**. Western blot analyses showing Sirt1-3 catalyzed decrotonylation of extracted histones on membrane.

**Figure supplement 2**. Analysis of the decrotonylation activities of Sirt1 and Sirt2 in cells.

**Figure supplement 3**. Subcellular locolization of Sirt3.

---

significantly upregulated on Sirt3 knockdown (*Figure 5F*). Given that histone Kcr is enriched at active gene promoters and potential enhancers (*Tan et al., 2011*), this positive correlation between the gene transcription level and the nearby histone Kcr level on Sirt3 knockdown suggests that Sirt3 might relieve a repressive effect on these target genes through 'erasing' histone Kcr 'marks'.

## Discussion

We have established a robust chemical proteomics approach to comprehensively profile histone decrotonylases. There have been important advances in our ability to detect PTMs. However, we currently lack reliable methods to identify, without bias, enzymes that regulate the addition and removal of PTMs, as interactions between PTMs and their regulating enzymes can be weak and transient, thereby limiting the applicability of conventional biochemical 'pull-down' methods. Our CLASPI approach overcame this difficulty by applying photo-cross-linking chemistry to convert weak and transient enzyme–PTM interactions into irreversible covalent linkages, and enabled a systematic profiling of the 'erasers' of protein PTMs. The present study has also largely broadened the scope of CLASPI technology from finding PTM 'readers'(*Li et al., 2012*; *Li and Kapoor, 2010*), which are usually involved in relatively more stable protein–protein interactions, to identifying dynamic and transient interactions between PTMs and their 'erasers'. We anticipate that this approach can be used to comprehensively profile 'erasers' of other PTMs, such as arginine demethylases.

Siruins were initially recognized as NAD-dependent deacetylases (*Imai et al., 2000*; *Landry et al., 2000*; *Sauve et al., 2006*). However, emerging evidence revealed that some sirtuins that displayed weak deacetylation activity had substrate specificity towards other acyl groups attached to lysine residues. For examples, Lin et al. recently demonstrated that Sirt5 can preferentially hydrolyze malonyl and succinyl lysine (*Du et al., 2011*; *Peng et al., 2011*), and Sirt6 can remove long chain fatty acyl groups (e.g., myristoyl group) from lysine residues (*Jiang et al., 2013*). In this study, we demonstrated that the three human sirtuins, Sirt1–Sirt3, catalyzed the hydrolysis of crotonyl lysine. This newly discovered decrotonylase activity broadens the landscape of PTMs that are targeted by sirtuins, and it also provides new impetus to investigate the cellular mechanisms and functions of Sirt1–Sirt3, which to date have been considered solely as deacetylases. This finding is also partially in agreement with the work of Denu and coworkers, in which only Sirt1 and Sirt2 exhibited decrotonylase activity, whereas Sirt3 was totally inactive, towards a histone H3K9Cr peptide in their radioactive [$^{32}$P]-NAD thin layer chromatography assay. In contrast, Sirt3 displayed robust decrotonylase activity against a variety of crotonylated histone peptides, including an H3K9Cr peptide in this study (*Figure 4*). Given the fact that the activity of Sirt3 can be peptide sequence-dependent (*Figure 4*), this discrepancy may be caused by the different H3K9Cr peptide substrates used in Denu's and this study, which consisted of amino acid residues 5–13 and 1–15 of histone H3, respectively.

We have demonstrated that endogenous Sirt3 functions as an 'eraser' to regulate histone crotonylation in cells. This finding opens new opportunities to investigate the cellular mechanisms and functions of histone crotonylation. In contrast, while the knockdowns of Sirt1 and Sirt2 did not cause accumulation of histone global or H3K4 crotonylation (*Figure 5—figure supplement 2*), we cannot rule out the possibility that these two sirtuins could target other histone crotonylation sites. Future studies are therefore needed to systematically profile mammalian crotonylome and analyze the lysine crotonylation sites that are targeted by Sirt1, Sirt2, and Sirt3, by comparing the corresponding wild-type and genetic knockout cells or tissues in conjunction with quantitative proteomics approaches.

The seven human sirtuins have distinct subcellular localizations. Sirt1, Sirt6, and Sirt7 are in the nucleus, Sirt3–Sirt5 localize to the mitochondria, and Sirt2 is primarily found in the cytoplasm (*Houtkooper et al., 2012*). However, Sirt3, in its full length form, has recently been found in the nucleus, and nuclear Sirt3 can associate with chromatin and result in repression of nearby genes (*Scher et al., 2007*; *Iwahara et al., 2012*). Based on the focused analysis at several Sirt3 target gene loci, the current study suggests a potential correlation of the transcriptional upregulation and the increase in local histone Kcr levels on Sirt3 knockdown. It also generates a hypothesis that Sirt3 could lead to silencing through 'erasing' Kcr at target genes. To test this hypothesis and examine the correlation between Sirt3 catalyzed histone deacrotonylation and gene expression genome-wide requires comprehensive profiling of global histone Kcr and gene expression regulated by Sirt3 using ChIP coupled to high throughput sequencing, in combination with RNA sequencing in future studies.

In addition, the same type of PTM at different modification sites of histones may have distinct effects on gene expression. For example, trimethylation at histone H3 Lys-4 (H3K4Me3) 'marks' genes that are being actively transcribed, whereas the same modification at H3 Lys-27 (H3K27Me3) 'marks' transcriptionally silent chromatin (*Martin and Zhang, 2005*). By analogy, it is possible that crotonylation at specific lysine sites of histones could also play different roles in the regulation of gene expression. This possibility may account for the fact that the transcription of the two genes (i.e., *Brip1* and *Zfat*) with elevated Kcr levels was not influenced in our study. The study of the effects of site-specific histone Kcr 'marks' (e.g., H3K4Cr) targeted by Sirt3 on the regulation of gene expression is an important next step.

## Materials and methods

### Reagents

Unless otherwise noted, all chemical reagents were purchased from Sigma–Aldrich (St. Louis, MO). Dulbecco's Modified Eagle Medium (DMEM) was purchased from Life Technologies. Ethylene diamine tetraacetic acid (EDTA) free protease inhibitor was purchased from Roche Applied Science (Germany). Pre-stained protein ladder was purchased from Bio-Rad (Hercules, CA). Pre-cast polyacrylamide gels (4–12% NuPAGE Bis-Tris gels) were purchased from Life Technologies. Mass spectrometry grade trypsin was purchased from Promega (Madison, WI). High capacity streptavidin beads were purchased from ThermoFisher Scientific (Waltham, MA). Antibodies were purchased from Santa Cruz Biotechnologies (Santa Cruz, CA) (anti-Sirt1, anti-Sirt2, and anti-γ-actin antibodies), Cell Signaling Technology (Danvers, MA) (anti-Sirt3, anti-HSP60, anti-fibrillarin, and anti-histone H3 antibodies), Abcam (United Kingdom) (anti-H3K4Ac and anti-H3K4Me3 antibodies), or PTM BioLabs (Chicago, IL) (anti-H3K4Cr, anti-H3K27Cr, and pan anti-crotonyllysine antibodies). Anti-Sirt3 N-term antibody was a generous gift from Dr Danny Reinberg (New York University, New York, United States).

### Instrumentation

In-gel fluorescence scanning was performed using a Typhoon 9410 variable mode imager (excitation 532 nm, emission 580 nm). Isothermal titration calorimetry measurements were performed on a MicroCal iTC200 titration calorimeter (Malvern Instruments, United Kingdom). Peptides were purified on a preparative HPLC system with Waters (Milford, MA) 2535 Quaternary Gradient Module, Waters 515 HPLC pump, Waters SFO System Fluidics Organizer, and Waters 2767 Sample Manager. Enzymatic reactions were monitored by an LC-MS system with Waters 1525 Binary HPLC Pump, Waters 2998 Photodiode Array Detector, and Waters 3100 Mass Detector. Detection of *O*-Cr-ADPR was carried out by Agilent (Santa Clara, CA) 1260 Infinity HPLC system connected to a Thermo Fisher Scientific LCQ DecaXP MS detector.

### Peptide synthesis and purification

All peptides were synthesized on Rink-Amide MBHA resin following a standard Fmoc based solid phase peptide synthesis protocol. Removal of protecting groups and cleavage of peptides from the

resin were done by incubating the resin with a cleavage cocktail containing 95% trifluoroacetic acid (TFA), 2.5% triisopropylsilane, 1.5% water, and 1% thioanisole for 2 hr. Peptides were purified by preparative HPLC with an XBridge Prep OBD C18 column (30 mm×250 mm, 10 μm; Waters). Mobile phases used were water with 0.1% TFA (buffer A) and 90% acetonitrile (ACN) in water with 0.1% TFA (buffer B). Peptides containing photo-cross-linker (benzophenone) were eluted with gradient 15–40% buffer B in 40 min; all other peptides were eluted with gradient 5–35% buffer B in 40 min. The elution rate was 15 mL/min. The purity and identity of the peptides were verified by LC-MS.

## Cell culture

HeLa S3, HEK293T, and HeLa cells were cultured in DMEM supplemented with 10% fetal bovine serum (FBS), 100 U/mL penicillin, and 100 μg/mL streptomycin. Cells were maintained in a humidified 37 °C incubator with 5% $CO_2$.

## Stable isotope labeling of amino acids in cell culture

HeLa S3 cells were grown in suspension at 37°C in a humidified atmosphere with 5% $CO_2$ in DMEM medium (–Arg, –Lys; Life Technologies) containing 10% dialyzed fetal bovine serum (Life Technologies), penicillin–streptomycin, and supplemented with 22 mg/L $^{13}C_6{}^{15}N_4$-L-arginine (Cambridge Isotope Laboratories, Tewksbury, MA) and 50 mg/L $^{13}C_6{}^{15}N_2$-L-lysine (Cambridge Isotope) or the corresponding non-labeled amino acids (Peptide International, Louisville, KY). Harvested cell pellets were washed with ice cold phosphate buffered saline (PBS) and frozen in liquid $N_2$. The cell powder grinded with a Ball Mill (Retch MM301) was stored at −80 °C until use.

## Preparation of whole-cell lysates for CLASPI experiment

To prepare whole-cell lysates, the frozen cell powder was first resuspended in a hypotonic buffer (10 mM HEPES, pH 7.5, 2 mM $MgCl_2$, 0.1% Tween-20, 20% glycerol, 2 mM phenylmethylsulfonyl fluoride (PMSF), and Roche Complete EDTA free protease inhibitors) and incubated for 10 min at 4 °C. The suspension was centrifuged at 16,000×g for 15 min at 4 °C and the supernatant was kept for use later. The pellet was resuspended in a high salt buffer (50 mM HEPES, pH 7.5, 420 mM NaCl, 2 mM $MgCl_2$, 0.1% Tween-20, 20% glycerol, 2 mM PMSF, and Roche Complete EDTA free protease inhibitors) and incubated for 30 min at 4 °C. The suspension was centrifuged at 16,000×g for 15 min at 4 °C, and the supernatant was combined with the soluble fraction in hypotonic buffer to give the whole-cell lysates.

## CLASPI photo-cross-linking

In a 'selectivity filter' experiment, probe **1** and probe **C** were incubated with heavy and light SILAC whole-cell lysates, respectively, in the binding buffer (50 mM HEPES, pH 7.5, 168 mM NaCl, 2 mM $MgCl_2$, 0.1% Tween-20, 20% glycerol, 2 mM PMSF, and Roche Complete EDTA free protease inhibitor cocktail) for 15 min at 4 °C. The samples were then irradiated at 365 nm using a Spectroline ML-3500S UV lamp for 15 min on ice. In 'an affinity filter experiment', the heavy and light SILAC lysates were reacted with probe **1** in the absence and presence, respectively, of H3K4Cr (1–15) peptide (30 μM) as a competitor. After photo-cross-linking, the heavy and light lysates were pooled.

## Cu(I)-catalyzed cycloaddition/click chemistry

Briefly, to the prepared samples, 100 μM of rhodamine azide for in-gel fluorescence scanning or cleavable biotin-azide for streptavidin enrichment were added, followed by 1 mM tris(2-carboxyethyl)phosphine and 100 μM tris[(1-benzyl-1H-1,2,3-triazol-4-yl)methyl]amine, and the reactions were initiated by the addition of 1 mM $CuSO_4$. The reactions were incubated for 1.5 hr at room temperature.

## Streptavidin affinity enrichment of biotinylated proteins

After the click chemistry with cleavable biotin-azide, the reaction was quenched by adding 4 volumes of ice cold acetone to precipitate the proteins. After washing with ice cold methanol twice, the air dried protein pellet was dissolved in PBS with 4% SDS, 20 mM EDTA, and 10% glycerol by vortexting and heating. The solution was then diluted with PBS to give a final concentration of SDS of 0.5%. High capacity streptavidin agarose beads (Thermo Fisher Scientific) were added to bind the biotinylated proteins with rotating for 1.5 hr at room temperature. To remove non-specific binding, the beads were washed with PBS with 0.2% SDS, 6 M urea in PBS with 0.1% SDS, and 250 mM $NH_4HCO_3$ with 0.05% SDS. The enriched proteins were then eluted by incubating with 25 mM $Na_2S_2O_4$, 250 mM $NH_4HCO_3$, and 0.05% SDS for 1 hr. The eluted proteins were dried down with SpeedVac.

## Sample preparation for mass spectrometry

The dried proteins were resuspended in 30 µL of lithium dodecyl sulfate sample loading buffer (Life Technologies) with 50 mM dithiothreitol (DTT), heated at 75 °C for 8 min, and then reacted with iodoacetamide in the dark for 30 min to alkylate all of the reduced cysteines. Proteins were then separated on a Bis-Tris gel, followed by fixation in a 50% methanol/7% acetic acid solution. The gel was stained by GelCode Blue stain (Pierce). The diced 1 mm (*Goldberg et al., 2007*) cubes of gels were then destained by incubating with 50 mM ammonium bicarbonate/50% acetonitrile for 1 hr. The destained gel cubes were dehydrated in acetonitrile for 10 min and rehydrated in 25 mM $NH_4HCO_3$ with trypsin for protein digestion at 37 °C overnight. The resulting peptides were enriched with StageTips. The peptides eluted from the StageTips were dried down by SpeedVac and then resuspended in 0.5% acetic acid for analysis by LC-MS/MS.

## Mass spectrometry

Mass spectrometry was performed on an LTQ-Orbitrap Velos mass spectrometer (Thermo Fisher Scientific). First, peptide samples in 0.1% formic acid were pressure loaded onto a self-packed PicoTip column (New Objective, Woburn, MA) (360 µm od, 75 µm id, 15 µm tip), packed with 7–10 cm of reverse phase C18 material (ODS-A C18 5-µm beads from YMC America, Allentown, PA), rinsed for 5 min with 0.1% formic acid, and subsequently eluted with a linear gradient from 2% to 35% B for 150 min (A=0.1% formic acid, B=0.1% formic acid in ACN, flow rate ~200 nL/min) into the mass spectrometer. The instrument was operated in a data-dependent mode, cycling through a full scan (300–2000 *m/z*, single µscan) followed by 10 CID MS/MS scans on the 10 most abundant ions from the immediate preceding full scan. Cations were isolated with a 2 Da mass window and set on a dynamic exclusion list for 60 s after they were first selected for MS/MS. The raw data were processed and analyzed using MaxQuant (version 1.2.2.5). A human fasta file (ipi.HUMAN.v.3.68.fasta) was used as the protein sequence searching database. Default parameters were adapted for protein identification and quantification. In particular, parent peak MS tolerance was 6 ppm, MS/MS tolerance was 0.5 Da, minimum peptide length was 6 amino acids, and maximum number of missed cleavages was 2. The proteins quantified were supported by at least two quantification events. Both the 'selectivity filter' and 'affinity filter' experiments were repeated twice, and only the proteins that were identified and quantified in all experiments were reported.

## In-gel fluorescence visualization

The click chemistry reactions were quenched by adding 1 volume of 2×sample buffer. The proteins were heated at 85 °C for 8 min, and resolved by SDS-PAGE. The labeled proteins were visualized by scanning the gel on a Typhoon 9410 variable mode imager (excitation 532 nm, emission 580 nm).

## Expression and purification of recombinant human sirtuins

Plasmids of Sirt1 (193–747), Sirt2 (36–356), Sirt5 (34–302), and Sirt6 (1–314) for Escherichia *coli* expression were generated as previously described (*Finnin et al., 2001*; *Du et al., 2011*; *Hubbard et al., 2013*; *Jiang et al., 2013*). Plasmids of Sirt3 (102–399) cloned in pTrcHis 2C vector for *E. coli* expression and full length Sirt3 (wide-type and mutant H248Y) cloned into pcDNA3.1 vector for mammalian cell expression were generous gifts from Dr Eric Verdin (University of California, San Francisco). Sirt3 mutant F180L was generated by site directed mutagenesis. All of the proteins were expressed in *E. coli* Rosetta cells. To induce expression of target proteins, isopropyl β-D-1-thiogalactopyranoside was added to a final concentration of 0.2 mM when $OD_{600}$ reached 0.6, and the culture was grown at 15 °C (Sirt3 at 25 °C) for 16–18 hr. Cells were harvested and resuspended in lysis buffer A (50 mM Tris–HCl, pH 7.5, 500 mM NaCl, 1 mM PMSF, and Roche EDTA free protease inhibitors, for Sirt1, Sirt2, and Sirt6) or buffer B (50 mM Tris–HCl, pH 7.5, 150 mM NaCl, 1 mM PMSF, and Roche EDTA free protease inhibitors, for Sirt3 and Sirt5). Following sonication and centrifugation, the supernatant was loaded onto a nickel column pre-equilibrated with lysis buffer. The column was washed with 5 column volumes of wash buffer (lysis buffer with 30 mM imidazole) and then the target proteins were eluted with elution buffer (lysis buffer with 250 mM imidazole). After purification, Sirt2 was digested by UPL1 at 4 °C overnight and purified by a Highload 26/60 Superdex75 gel filtration column (GE Healthcare Life Sciences, United Kingdom). Sirt6 was purified by SP column and Superdex75 gel filtration column. Others were loaded onto a Superdex75 gel filtration or Highload 26/60 Superdex200 (for Sirt1) column. After concentration, the target proteins were frozen and stored at −80 °C.

## Isothermal titration calorimetry measurements

Experiments were performed at 25 °C on a MicroCal iTC200 titration calorimeter (Malvern Instruments). The reaction cell containing 200 µL of 100–200 µM proteins was titrated with 17 injections (firstly 0.5 µL, and all subsequent injections 2 µL of 1.5–2.5 mM peptides). The binding isotherm was fit with Origin 7.0 software package (OriginLab, Northampton, MA) that uses a single set of independent sites to determine the thermodynamic binding constants and stoichiometry.

## Crystallization, X-ray data collection, and structure determination

Sirt3/H3K4Cr mixtures were prepared at a 1:20 protein/peptide molar ratio and incubated for 60 min on ice. Crystals of Sirt3 (102–399) complexed with H3K4Cr (1–10) peptide were obtained by the hanging drop vapor diffusion method at 291 K using commercial screens from Hampton Research (Aliso Viejo, CA). Each drop, consisting of 1 µL of 10 mg/mL protein complex solution (20 mM Tris–HCl, pH 7.4, 100 mM NaCl, and 5 mM DTT) and 1 µL of reservoir solution, was equilibrated against 400 µL of reservoir solution. The qualified crystals of Sirt3 grew with a cube profile within 1 week with a reservoir containing 12% PEG4K, 0.1 M sodium malonate, pH 6.5, and 5% isopropanol. The mixture of 25% glycerol with the reservoir solution above was used as the cryogenic liquor. The X-ray diffraction data were collected at 100 K in a liquid nitrogen gas stream using the Shanghai Synchrotron Radiation Facility beamline 17U ($\lambda$ = 0.9791 Å). A total of 120 frames were collected with a 1° oscillation and the data were indexed and integrated using the program HKL2000 (*Otwinowski and Minor, 1997*). The complex structure of Sirt3 with H3K4Cr peptide was solved by molecular replacement using the program Molrep from the CCP4 Suit (*Collaborative Computational Project, Number 4, 1994*), with the published Sirt3 structure (PDB: 3GLR) (*Jin et al., 2009*) as the search model. Refinement and model building were performed with REFMAC5 and COOT from CCP4. The X-ray diffraction data collection and structure refinement statistics are shown in *Supplementary file 1*.

## Enzymatic reactions

The enzymatic activities of human sirtuins were measured by detecting the removal of the crotonyl group from peptides (*Du et al., 2011*). Sirtuin protein (5 µM) was incubated with 500 µM of corresponding crotonylated peptides and 1 mM of NAD in a reaction buffer containing 20 mM Tris–HCl buffer (pH 7.5) and 1 mM DTT at 37 °C for 2 hr. The reactions were stopped by adding one-third reaction volume of 20% TFA and immediately frozen in liquid $N_2$. For Sirt3, samples without NAD or without enzyme were treated under the same conditions as the controls. Samples were then analyzed by LC-MS with a Vydac 218TP C18 column (4.6 mm×250 mm, 5 µm; Grace Davison, Columbia, MD). Mobile phases used were 0.05% TFA in water (buffer A) and 0.05% TFA in ACN (buffer B). The flow rate for LC was 0.6 mL/min. The peptide mixtures were eluted by buffer A for 10 min and then 0–30% buffer B over 10 min. MS started to record at 10 min for each injection.

## Determination of $k_{cat}$ and $K_m$

Enzyme was incubated with different concentrations of corresponding peptides bearing two tryptophans at the C terminus (20, 40, 60, 80, 100, 200, and 500 µM) and 1.0 mM NAD in 20 mM Tris–HCl buffer (pH 7.5) containing 1 mM DTT in 25 µL reaction at 37°C for a certain period of time within the initial linear range. The enzyme concentration and reaction time used were: Sirt3–H3K4Ac: 1 µM enzyme, 5 min; Sirt3–H3K4Cr: 1 µM enzyme, 20 min; Sirt3 (F180L)–H3K4Ac: 0.8 µM enzyme, 5 min; and Sirt3 (F180L)–H3K4Cr: 5 µM enzyme, 20 min. The reactions were stopped by adding one-third reaction volume of 20% TFA and immediately frozen in liquid $N_2$. Samples were then analyzed by HPLC with a Vydac 218TP C18 column (4.6 mm×250 mm, 5 µm; Grace Davison). Mobile phases used were water with 0.1% TFA (buffer A) and 90% ACN in water with 0.1% TFA (buffer B). The wavelength for UV detection was 280 nm. The analysis gradient for deacetylation samples was 16% buffer B for 20 min with a flow rate at 1.5 mL/min. The analysis gradient for decrotonylation samples was 15–35% buffer B in 12 min with a flow rate at 1.0 mL/min.

## Detection of *O*-Cr-ADPR

Sirt3 (5 µM) was incubated with 500 µM of H3K4Cr (1–15) peptide and 1 mM of NAD in a reaction buffer containing 20 mM Tris–HCl buffer (pH 7.5) and 1 mM DTT at 37°C for 2 hr. The reactions were stopped by immediately freezing in liquid $N_2$. Sample was then analyzed by LC-MS with a VisionHT C18 column (2.1 mm×150 mm, 3 µm; Grace Davison) on an Agilent 1260 Infinity HPLC system, followed by Thermo Fisher Scientific LCQ DecaXP MS Detector. Mobile phases used were 0.02% TFA in

water (buffer A) and 90% ACN in water with 0.02% TFA (buffer B). The flow rate for LC was 0.2 mL/min. The sample was eluted by buffer A for 10 min and then 0–10% buffer B over 10 min. The wavelength for UV detection was 260 nm. MS started to record at 10 min.

## RNAi experiments

Sirt1 siRNA 15 nM (Santa Cruz Biotechnologies), Sirt2 siRNA 30 nM (Thermo Fisher Scientific), or Sirt3 siRNA 30 nM (Thermo Fisher Scientific) was transfected into a HeLa cell line with DharmaFECT 1 Transfection Reagent (Thermo Fisher Scientific), according to the manufacturer's instructions. Corresponding concentrations of control siRNA were used as negative controls. Following transfection, cells were then maintained in a humidified 37 °C incubator with 5% $CO_2$ for another 48 hr (for Sirt1 and Sirt2) or 72 hr (for Sirt3).

## Histone extraction

An acid extraction method was used to isolate histones from HeLa S3 cells (*Shechter et al., 2007*). Briefly, the harvested HeLa S3 cell pellet was resuspended with lysis buffer (10 mM Tris–HCl pH 8.0, 1 mM KCl, 1.5 mM $MgCl_2$, 1 mM DTT 2 mM PMSF, and Roche Complete EDTA free protease inhibitors) and incubated at 4 °C by rotating for 1 hr. The intact nuclei were pelleted by centrifuging at 10,000×g for 10 min at 4 °C. To extract histones, 0.4 N $H_2SO_4$ was added to resuspend the nuclei, followed by rotating at 4°C overnight. After centrifuging to remove the nuclei debris, histones were precipitated by adding 100% trichloroacetic acid drop by drop (trichloroacetic acid final concentration 33%). The precipitated histones were pelleted at 16,000×g for 10 min at 4 °C and washed with ice cold acetone twice. The air dried protein pellet was dissolved with dd$H_2O$ and stored at −80 °C for later use.

## On-membrane decrotonylation experiment

HeLa S3 whole-cell lysate (20 µg) or 5 µg of extracted histones were resolved by SDS-PAGE gel and transferred to PVDF membranes. The membranes were incubated with or without 0.1 µM of Sirt3 in reaction buffer (25 mM Tris–HCl, 130 mM NaCl, 3 mM KCl, 1 mM $MgCl_2$, and 1 mM DTT, pH 7.5) containing 1 mM NAD at 37 °C for 2 hr.

## In-solution decrotonylation experiment

Extracted histones (4 µg) were incubated with or without 1 µM or 5 µM of Sirt3 in reaction buffer (25 mM Tris–HCl, 130 mM NaCl, 3 mM KCl, 1 mM $MgCl_2$, and 1 mM DTT, pH 7.5) containing 1 mM NAD at 37 °C overnight.

## Immunofluorescence

HeLa cells grown on coverslips were fixed with 3.7% polyformaldehyde in PBS, permeabilized with 0.1% Triton X-100 in PBS, and blocked for 30 min at room temperature using 5% bovine serum albumin (dissolved with PBS containing 0.1% Triton X-100). Cells were incubated with primary antibody overnight at 4°C and washed trice with PBST (0.1% Tween-20 in PBS) prior to secondary antibody (containing DAPI for nucleus staining) incubation at room temperature for 1 hr. Washed cells were then subjected to a Zeiss LSM 510 laser scanning confocal microscope.

## Subcellular fractionation

In brief, HeLa cells were harvested by centrifugation and washed with PBS twice; all subsequent steps were performed at 4 °C. Cells were then suspended in 5 cell pellet volumes of buffer A (10 mM HEPES, pH 7.9 at 4 °C, 1.5 mM $MgCl_2$, 10 mM KCl, and 0.5 mM DTT) followed by incubation for 10 min. After centrifugation, cells were resuspended in 2 cell pellet volumes of buffer A and lysed by Dounce homogenizer (B type pestle) with homogenate checked by microscopy. The cell lysis was layered over 30% sucrose in buffer A and then centrifuged for 15 min at 800×g. The resulting pellet was recovered from the sucrose phase, washed by buffer A twice, and then extracted by buffer C (20 mM HEPES, pH 7.9, 25% (vol/vol) glycerol, 0.42 M NaCl, 1.5 mM $MgCl_2$, 0.2 mM EDTA, 0.5 mM PMSF, and 0.5 mM DTT) for 30 min at 4 °C. After centrifugation at 12,000×g for 30 min, the supernatant was termed the nuclear fraction. The resulting supernatant was centrifuged twice at 800×g to complete the pellet nuclei and intact cell. The supernatant was then centrifuged at 7,000×g to pellet the mitochondria followed by washing twice with buffer A. The mitochondria were then lysed by TXIP-1 buffer (1% Triton X-100 (vol/vol), 150 mM NaCl, 0.5 mM EDTA, and 50 mM Tris–HCl, pH 7.4). Protein concentration was determined by BCA assay.

## Immunoblotting

Proteins separated by SDS-PAGE were transferred onto a PVDF membrane which was then blocked (5% non-fat dried milk and 0.1% Tween-20 in PBS) for 1 hr at room temperature. The membrane was incubated with primary antibody diluted in PBST with 2% bovine serum albumin, followed by washing with PBST for 5 min trice, incubated with goat anti-rabbit-horseradish peroxidase conjugated secondary antibody (1:20000; Santa Cruz Biotechnologies), or rabbit anti-mouse- horseradish peroxidase conjugated secondary antibody (1:5000; Santa Cruz Biotechnologies) diluted in PBST for 1 hr at room temperature, and then visualized with western blotting detection reagents (Thermo Fisher Scientific).

## Gene expression analysis

Total RNA was isolated using TRIzol Reagent (Life Technologies). RNA was reverse transcribed into cDNA by M-MLV Reverse Transcriptase (Life Technologies) using oligo (dT) primers. qPCR was performed using Power SYBR Green PCR Master Mix (Life Technologies) on an ABI StepOnePlus system following the manual's instructions. All primers used are listed in *Supplementary file 2*.

## ChIP and qPCR

Cells were cross linked by 1% formaldehyde for 10 min and quenched by 0.125 M glycine for 5 min at room temperature. Cells were then lysed by ChIP lysis buffer (5 mM PIPES pH 8.0, 85 mM KCl, and 1% IGEPAL CA-630) and homogenized using a glass Dounce homogenizer (type B pestle). The nuclear fraction was precipitated and lysed in nuclei lysis buffer (50 mM Tris–HCl, pH 8.0, 10 mM EDTA, and 1% SDS) for 30 min at 4 °C. The nuclear lysis was sonicated to a chromatin ranging from 600 bp to 800 bp. Immunoprecipitation was done in immunoprecipitation dilution buffer (50 mM Tris–HCl, pH 7.4, 150 mM NaCl, 1% IGEPAL CA-630, 0.25% deoxycholic acid, and 1 mM EDTA) using Dynabeads coupled with Protein G (Life Technologies). Chromatin (5 µg) and 8 µg of pan anti-crotonyllysine antibody were used for each ChIP reaction. Chromatin complex was eluted from beads by ChIP elution buffer (50 mM $NaHCO_3$ and 1% SDS) and added to 5 M NaCl to a final concentration of 0.54 M. To reverse cross links of protein/DNA complex to free DNA, samples were incubated at 65 °C for 2 hr followed by 95 °C for 15 min. After incubation with RNase (Thermo Fisher Scientific) for 20 min at 37 °C, DNA was recovered and used for qPCR, as described above. All primers used are listed in *Supplementary file 2*.

## Acknowledgements

This work was supported in part by the Hong Kong Research Grants Council (RGC) Early Career Scheme (ECS) (HKU 709813P) and the General Research Fund (GRF17303114) to XDL, GRF766510 to QH, and Hung Hing Ying Physical Science Research Fund (20373739) to XDL. We thank C-M Che and YME Fung for support on the mass spectrometer, and staff at the Shanghai Synchrotron Radiation Facility for assistance during data collection. We thank D Reinberg for anti-Sirt3 N-term antibody.

## Additional information

### Funding

| Funder | Grant reference number | Author |
| --- | --- | --- |
| Research Grants Council, University Grants Committee, Hong Kong | Early Career Scheme, HKU 709813P | Xiang David Li |
| Research Grants Council, University Grants Committee, Hong Kong | General Research Fund, GRF766510 | Quan Hao |
| University of Hong Kong | Hung Hing Ying Physical Science Research Fund, 20373739 | Xiang David Li |
| Research Grants Council, University Grants Committee, Hong Kong | General Research Fund, GRF17303114 | Xiang David Li |

The funders had no role in study design, data collection and interpretation, or the decision to submit the work for publication.

## Author contributions

XB, XL, Acquisition of data, Analysis and interpretation of data, Contributed unpublished essential data or reagents; YW, X-ML, CFW, Acquisition of data, Analysis and interpretation of data; ZL, TY, Acquisition of data, Contributed unpublished essential data or reagents; JZ, Conception and design, Analysis and interpretation of data; QH, Conception and design, Analysis and interpretation of data, Drafting or revising the article; XDL, Conception and design, Acquisition of data, Analysis and interpretation of data, Drafting or revising the article

## Additional files

### Supplementary files

• Supplementary file 1. Diffraction data and structure refinement statistics.

• Supplementary file 2. Primers information.

### Major dataset

The following previously published dataset was used:

| Author(s) | Year | Dataset title | Dataset ID and/or URL | Database, license, and accessibility information |
|---|---|---|---|---|
| Jin L, Wei W, Jiang Y, Peng H, Cai J, Mao C, Dai H, Choy W, Bemis JE, Jirousek MR, Milne JC, Westphal CH, Perni RB | 2009 | Crystal structure of human SIRT3 with acetyl-lysine AceCS2 peptide | http://www.pdb.org/pdb/explore/explore.do?structureId=3GLR | Publicly available at RCSB Protein Data Bank. |

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
