## [Decision Letter]

Thank you for sending your work entitled “Identification of 'erasers' for lysine-crotonylated histone marks using a chemical proteomics approach” for consideration at *eLife*. Your article has been evaluated by Michael Marletta (Senior editor), Wilfred van der Donk (Reviewing editor), and 3 reviewers. The study was judged to be of interest and could be suitable for *eLife* provided you are able to fully address some critical questions that were raised.

The Reviewing editor and the other reviewers discussed their comments before we reached this decision, and the Reviewing editor has assembled the comments below to help you prepare a revised submission. Overall, the reviewers found the observed decrotonylation activity interesting and of high potential impact, but they felt that some key and essential experiments are required to support the interpretation of the authors.

1) One of the main concerns is the localization of Sirt3. The authors cite references that suggest it could be in the nucleus, but other studies have reported exclusive localization to the mitochondria. Without conclusive evidence that active Sirt3 is indeed in the nucleus in the experiments performed, the relevance of histone decrotonylation would be in question.

2) A closely related question is the impact of Sirt1-3 knockdown/overexpression on histone acetylation and methylation (1, 2 and 3 Me) levels of H3K4. For instance, could the observed effects on crotonylation be caused by changes of other modifications at these sites in these proteins due to knockdown? These experiments would be straightforward and are critical to support the interpretation of the authors.

3) A different important concern is the lack of quantitation with respect to the decrotonylation activity. The authors convincingly demonstrate that decrotonylation occurs *in vitro* and that O-crotonyl-ADPR is formed, but without any steady-state kinetic parameters it is unclear how robust (and hence meaningful) this activity is. Only physiologically relevant kinetic parameters would support a functional role of histone decrotonylation.

4) The biological consequences (and therefore importance) of decrotonylation activity are currently not clear. Given the apparent selectivity for histones (based on Figure 5), a relatively straightforward first look would involve analysis of transcriptional changes upon knockdown of Sirt3. RNA-seq with H3K4Cr Chip-Seq, or alternatively focusing on a few loci and examining potential correlations of H3K4Cr (Chip-PCR) upon Sirt3 knockdown, would provide important information.

5) The structural studies of crotonylated peptide to SIRT3 propose a binding mode involving Phe180 that is highly conserved. What is not mentioned is that this residue is known to interact strongly with NAD in prior structural work (e.g. Wolbergers comprehensive studies, which include termolecular structures of both NAD and peptides bound to sirtuin active sites). The authors state that this residue is not conserved in SIRT4-7, but its identification and conservation has been supported by many sequence comparison and structural studies. The observed lack of decrotonylation activity upon mutation may therefore be a consequence of disrupted NAD binding rather than demonstrating substrate selectivity. It also questions whether the observed binding mode in the absence of NAD is relevant if NAD is bound to the enzyme. Steady state parameters for the decrotonylation activity of the mutant and binding studies of NAD and the crotonylated peptide to the mutant may be able to address these concerns.

---

## [Author Response]

*1) One of the main concerns is the localization of Sirt3. The authors cite references that suggest it could be in the nucleus, but other studies have reported exclusive localization to the mitochondria. Without conclusive evidence that active Sirt3 is indeed in the nucleus in the experiments performed, the relevance of histone decrotonylation would be in question*.

We appreciate this concern raised by the reviewers and also agree that the subcellular localization of Sirt3 has been debated. To address this issue, we would like to provide both a detailed literature review and our own experimental data below.

*Literature review*:

We first carefully reviewed the literature for the detailed experimental data that indicate the mitochondrial and nuclear localization of Sirt3, respectively. Here we summarize our findings in the table shown below.

**Table:** Summary of studies of Sirt3’s subcellular localization in the literature.Ref. No.Sirt3 LocalizationEvidenceFig. No. in Ref.Cell TypeAntibodyMethodsIBIFGFPOthers**1**MitochondriaOverexpressed Sirt3-FLAG was found in mitochondrial extraction.**2b**HEK293Tanti-FLAG✔MitochondriaEndogenous Sirt3 was detected in mitochondrial fraction with rabbit sirt3 antiserum (anti-sirt3 C term 15 AAs).**2c**HEK293Tanti-Sirt3 C-term✔MitochondriaOverexpressed Sirt3-GFP localized in mitochondria**2a**HeLa✔**2**MitochondriaOverexpressed Sirt3-EGFP was detected in mitochondria by immunoelectron microscopy.**6**Cervical carcinomaImmunoelectronMitochondriaOverexpressed full length Sirt3-EGFP was present in mitochondria**3, 4, 5**COS7✔**3**MitochondriaOverexpressed Sirt3-GFP was detected in mitochondria.**1a**NHF✔MitochondriaThere was no detectable signal of overexpressed Sirt3 with N-term-HA tag by immunofluorescence.**−**NHFanti-HA,✔MitochondriaOverexpressed of Sirt3-V5-His with C-term-tag was detected in mitochondria by immunofluorescence.**3a, b**NHF HeLaanti-V5✔MitochondriaSirt3-V5-His was detected in mitochondrial extraction using anti-V5 antibody.**3c**HEK293Tanti-V5✔**4**MitochondriaOverexpressed mSirt3-FLAG was detected in mitochondria by immunofluorescence.**2a**NIH3T3 (mouse)anti-FLAG✔**5**MitochondriaCo-expression of AceCS1/2 with Sirt3-FLAG down-regulated acetylation level on AceCS2 (in mitochondria), but not AceCS1 (in nucleus).**1e, 3c**COS7anti-FLAG✔**6**MitochondriaEndogenous Sirt3 was detected in mitochondrial extraction using anti-sirt3 C-term serum.**1b-d**HEK293anti-sirt3 C-term✔MitochondriaAceCS2-FLAG localized in mitochondria. Endogenous Sirt3 was co-precipitated from cell stably expressing AceCS2-FLAG using anti-FLAG antibody.**1a, 3e**HeLa HEK293anti-FLAG anti-sirt3 C-term✔MitochondriaOverexpression of Sirt3-FLAG down-regulated acetylation level on AceCS2.**3d**COS-1✔**7**MitochondriaOverexpressed Sirt3-myc was detected in mitochondria by immunofluorescence.**1a**COS7anti-Myc tag✔MitochondriaOverexpressed Sir3-Myc was detected in mitochondrial extraction using anti-Myc antibody.**1b**COS7anti-Myc tag✔**Nucleus**Co-expression of Sirt5-FLAG with Sirt3-Myc promoted translocation of Sirt3 into nucleus.**2a**COS7anti-Myc tag✔**Nucleus**Overexpressed Sirt3-Myc could be detected in total lysate but not in post-nuclear supernatant after co-expression with Sirt5-FLAG.**2b, c**COS7anti-Myc tag✔MTSSirt3-Myc with mitochondria targeting signal (72-75) mutant to 4 Ala was found in in both nucleus and cytoplasm.**3**COS7anti-Myc tag✔NTSSirt3-Myc with nuclear targeting signal (214-216) mutant to 3 Ala was detected in nucleus and cytoplasm when co-expressed with Sirt5.**4b, c**COS7anti-Myc tag✔**8**MitochondriaEndogenous Sirt3 was detected in mitochondria isolation using anti-Sirt3 C-term antibody.**1b-d**mouseanti-Sirt3 C-term✔**9**MitochondriaOverexpressed Sirt3-FLAG or Sirt3-Myc was detected in mitochondria by immunofluorescence.**1A**HeLa HEK293T U2OSanti-FLAG, anti-Myc✔MitochondriaLMB treatment did not result in nuclear translocation of Sirt3-FLAG but did for HDAC7.**1B, C**HeLa U2OSanti-FLAG, anti-Myc✔MitochondriaOverexpressed POLG2-HA, a positive control for mitochondrial localized proteins, was detected in mitochondria isolation, by not for Sirt3-FLAG.**2A**HEK293anti -Sirt3 SC49744✔**10****Nucleus**Full length Sirt3 was detected in all subcellular fractionation parts.**5**Cardiomyocytesanti-sirt3 Ab56214,✔**Nucleus**Endogenous Sirt3 was detected in nucleus by microscopy.**6a-e**Cardiomyocytesanti-sirt3 SC49744✔**Nucleus**Overexpressed Sirt3 was detected in nucleus by immunofluorescence.**6f**
**7a, b**Cardiomyocytesanti -Sirt3 SC49744✔**Nucleus**Full length Sirt3 was detected in nuclear extraction using anti-Sirt3 antibody.**7d, g**Cardiomyocytesanti-sirt3 SC49744✔**Nucleus**Ku70, a nuclear protein, could be co-precipitated by anti-Sirt3 antibody.**8a**Cardiomyocytesanti-Sirt3 AP6242a✔**Nucleus**Ku70 was co-precipitated with FLAG antibody from Sirt3-FLAG expressing cells but not from cells expressing the FLAG tag alone.**8b**COS7anti-FLAG,✔**Nucleus**Sirt3 was pulled down with Ku70 using anti-Ku70 antibody.**8c**Cardiomyocytesanti-sirt3 PAB11098✔**Nucleus**Precipitation of recombinant Ku70 by recombinant Sirt3 verified their direct interaction *in vitro.***8d**✔**Nucleus**Acetylation level on Ku70 was reduced by overexpressed Sirt3 and enhanced by Sirt3 knockdown via RNAi.**9d****10c, d**HeLa✔**Nucleus**Sirt3 was capable of deacetylation on Ku70 to enhance Ku70/Bax bindig to prevent cell death.**11, 12**HeLa✔**11****Nucleus**Endogenous full-length Sirt3 was detected in nucleus by immunofluorescence using anti-Sirt3 N-term antibody.**1b**HeLa & HEK293Tanti-sirt3 N-term✔**Nucleus**Co-transfection of Gal4-Sirt3 and luciferase reporter plasmid showed the nucleus localization of Gal4-Sirt3.**3b**HEK293Tanti-Gal-DBD✔**Nucleus**Full length Sirt3 resided in the nucleus and expelled from nucleus upon cellular stress.**5**HeLa S3anti-sirt3 N-term✔**Nucleus**The synthesis of tetracycline-inducible Sirt3 harboring a C-term HA-tag is processed in the nucleus.**6a**HEK293Tanti-HA✔**Nucleus**Overexpressed Sirt3-HA was detected in nuclear extraction.**6b**HEK293Tanti-HA✔**12****Nucleus**Stress-induced the degradation of full-length Sirt3 in nuclei.**5b, c**HeLa, U2OS, HEK293Tanti-HA anti-Sirt3 sc-49744✔**Nucleus**SIRT3 localizes to defined chromatin regions and regulates gene expression.**6, 7, 8**U2OS✔**Nucleus**Interaction of endogenous Sirt3 and SKP2, a protein found outside mitochondria, was validated by immunoprecipitation.**4**HEK293Tanti-HA anti-FLAG anti-Sirt3 C73E3✔

References:

1) Schwer B, North BJ, Frye RA, Ott M, Verdin E. The human silent information regulator (Sir)2 homologue hSIRT3 is a mitochondrial nicotinamide adenine dinucleotide–dependent deacetylase, *J Cell Biol*. 2002; 158(4):647-57

2) Onyango P, Celic I, McCaffery JM, Boeke JD, Feinberg AP. SIRT3, a human SIR2 homologue, is an NAD- dependent deacetylase localized to mitochondria, *Proc Natl Acad Sci*. 2002; 99(21):13653-8

3) Michishita E, Park JY, Burneskis JM, Barrett JC, Horikawa I. Evolutionarily Conserved and Nonconserved Cellular Localizations and Functions of Human SIRT Proteins, *Mol Biol Cell.* 2005; 16(10):4623-35

4) Shi T, Wang F, Stieren E, Tong Q. SIRT3, a Mitochondrial Sirtuin Deacetylase, Regulates Mitochondrial Function and Thermogenesis in Brown Adipocytes, *J Biol Chem*. 2005; 280(14):13560-7

5) Hallows WC, Lee S, Denu JM. Sirtuins deacetylate and activate mammalian acetyl-CoA synthetases, *Proc Natl Acad Sci*. 2006; 103(27):10230-5

6) Schwer B, Bunkenborg J, Verdin RO, Andersen JS, Verdin E. Reversible lysine acetylation controls the activity of the mitochondrial enzyme acetyl-CoA synthetase 2, *Proc Natl Acad Sci*. 2006; 103(27):10224-9

7) Nakamura Y, Ogura M, Tanaka D, Inagaki N. Localization of mouse mitochondrial SIRT proteins: Shift of SIRT3 to nucleus by co-expression with SIRT5, *Biochem Biophys Res Commun.* 2008; 366(1):174-9

8) Lombard DB *et al.* Mammalian Sir2 Homolog SIRT3 Regulates Global Mitochondrial Lysine Acetylation, *Mol Cell Biol.* 2007; 27(24):8807-14

9) Cooper HM, Spelbrink JN. The human SIRT3 protein deacetylase is exclusively mitochondrial, *Biochem J.* 2008; 411(2):279-85

10) Sundaresan NR, Samant SA, Pillai VB, Rajamohan SB, Gupta MP. SIRT3 Is a Stress-Responsive Deacetylase in Cardiomyocytes That Protects Cells from Stress-Mediated Cell Death by Deacetylation of Ku70, *Mol Cell Biol.* 2008;28(20):6384-401

11) Scher MB, Vaquero A, Reinberg D. SirT3 is a nuclear NAD+-dependent histone deacetylase that translocates to the mitochondria upon cellular stress, *Genes Dev*. 2007; 21(8):920-8

12) Iwahara T, Bonasio R, Narendra V, Reinberg D. SIRT3 Functions in the Nucleus in the Control of Stress-Related Gene Expression, *Mol Cell Biol*. 2012; 32(24):5022-34

From the analysis shown in this table, we found mainly two lines of evidences that suggest Sirt3 localizes exclusively in mitochondria. The first line of evidence involves the direct visualization of recombinant GFP-fused or epitope-tagged Sirt3 in the mitochondria of cells through GFP fluorescence or immunofluorescence (IF) using antibodies against the fused tags, respectively (see references 1-4 and 7-9 in the table above). The second line of evidence is from the detection of overexpressed epitope-tagged Sirt3 and endogenous Sirt3 in the mitochondrial fractions of cells by immunoblotting (IB) using anti-tag and anti-Sirt3 antibodies, respectively (see reference 1,3 and 6-9 in the table above). It should be noted that in these studies the detection of endogenous Sirt3 relied on antibodies that recognize the C-terminal regions of Sirt3. These anti-Sirt3 (*C*-term) antibodies have been used to reveal the localization of endogenous Sirt3, more specifically, short-form (i.e., N-terminal truncated) Sirt3, in the mitochondria fractions using immunoblotting analysis. However, the application of the anti-Sirt3 (*C*-term) antibodies to detect of endogenous full-length Sirt3 has not been limited in these studies, most likely due to the relatively low abundance of full-length Sirt3. Also, the IF detection of endogenous Sirt3 using these antibodies has NOT been shown in these studies.

In contrast, as shown in the table, several studies have provided evidences to support that while short-form Sirt3 predominantly localize in the mitochondria, full-length Sirt3 can be present in the nucleus.(need to check carefully about IB data) Reinberg *et al.* generated an antibody that specifically targets a Sirt3 N-terminal region. Using this anti-Sirt3 (*N*-term) antibody, endogenous Sirt3 was detected in the nucleus of HEK 293 and HeLa cells by IF (see reference 11 in the table above). Furthermore, a ChIP-Seq experiment using the anti-Sirt3 (*N*-term) antibody revealed the Sirt3’s localization on chromatin and the transcriptional level of several tested genes where Sirt3 localizes were indeed regulated by Sirt3 (see reference 12 in the table above).

Our experimental results:

To examine the subcellular localization of Sirt3, we also performed a couple of experiments by ourselves. First, we prepared mitochondrial and nuclear fractions from HeLa cells and examined whether endogenous Sirt3 were present in these fractions by Western blotting using antibodies that target the *C*-terminal (from Cell signaling) and the *N*-terminal (from D. Reinberg lab) regions of Sirt3, respectively. Consistent with the literature, while the short-form Sirt3 were identified in the mitochondrial fraction exclusively, we indeed detected full-length Sirt3 in the nuclear fraction (see Figure 5—figure supplement 3). Furthermore, using anti-Sirt3 (*N*-term) antibody, we were also able to show that Sirt3 indeed localize in the nucleus of the HeLa cells using IF (see Figure 5—figure supplement 3). In addition, another line of evidence that supports the nuclear localization of Sirt3 came from our chromatin immunoprecipitation (ChIP) type of experiments. As mentioned above, Sirt3 was found to localize to a broad variety of genomic regions (see reference 12 in the table above). We therefore focused some of these regions and examined whether Sirt3 could regulate the lysine crotonylation (Kcr) levels on these regions. Our data showed that knockdown of Sirt3 indeed caused significant increase in Kcr levels of the tested genomic regions (see Figure 5). This result will be further discussed in our response to the point #4 raised by the reviewers.

Taken together, although Sirt3 was identified as the primary deacetylase in mitochondria and involved in metabolic regulation through controlling protein acetylation dynamics, it did not rule out the possibility that Sirt3 may play roles outside mitochondria. Indeed, both the literature review and our experimental data suggest that Sirt3 can be present and function in nucleus.

*2) A closely related question is the impact of Sirt1-3 knockdown/overexpression on histone acetylation and methylation (1, 2 and 3 Me) levels of H3K4. For instance, could the observed effects on crotonylation be caused by changes of other modifications at these sites in these proteins due to knockdown? These experiments would be straightforward and are critical to support the interpretation of the authors*.

We appreciate the reviewers’ thoughtful comments. To rule out the possibility that the increase in the crotonylation level on histone H3K4 upon Sirt3 knockdown was caused by the changes of other modifications on the same site, we also examined the acetylation (H3K4Ac) and methylation (H3K4Me3) levels in the Sirt3 knockdown experiments. Our data showed that while the H3K4Cr level was significantly enhanced by Sirt3 knockdown, the H3K4Ac and H3K4Me3 levels remained unaltered, suggesting that Sirt3 indeed selectively targets histone crotonylation. We have now included this data in Figure 5.

*3) A different important concern is the lack of quantitation with respect to the decrotonylation activity. The authors convincingly demonstrate that decrotonylation occurs in vitro and that O-crotonyl-ADPR is formed, but without any steady-state kinetic parameters it is unclear how robust (and hence meaningful) this activity is. Only physiologically relevant kinetic parameters would support a functional role of histone decrotonylation*.

We agree with the reviewers that the steady-state kinetic parameters are important to this newly identified decrotonylase. In the revised manuscript (see Figure 5—figure supplement 3 in the revised manuscript), we determined the *k*_*cat*_ (0.01 s^-1^)*, K*_*m*_ (12.6 μM)*, and k*_*cat*_*/K*_*m*_ (7.83 × 10^2^ s^-1^M^-1^) values for Sirt3’s decrotonylase activity using H3K4Cr as the substrate. These values are comparable to the steady-state kinetic parameters for some recently identified deacylases, such as Sirt5 as demalonylase and desuccinylase, and Sirt6 as de-long-chain fatty acylase (e.g., demyristoylase). Therefore, we believe the decrotonylase activity of Sirt3 is robust.

*4) The biological consequences (and therefore importance) of decrotonylation activity are currently not clear. Given the apparent selectivity for histones (based on*
Figure 5*), a relatively straightforward first look would involve analysis of transcriptional changes upon knockdown of Sirt3. RNA-seq with H3K4Cr Chip-Seq, or alternatively focusing on a few loci and examining potential correlations of H3K4Cr (Chip-PCR) upon Sirt3 knockdown, would provide important information*.

We appreciate the reviewers’ helpful comments and suggestions about the preliminary exploration of potential biological significance of (de)crotonylation. It has been reported that Sirt3 can bind to chromatin and cause the repression of the neighboring genes in U2OS cells. We therefore hypothesized that Sirt3 could regulate the gene transcription via controlling the local histone Kcr levels. To test this hypothesis, we focused on seven candidate genes, *Baz2a*, *Brip1*, *Corin*, *Ptk2*, *Tshz3*, *Wapal* and *Zfat*, whose transcription start sites (TSS) are close to Sirt3 enriched region. Chromatin precipitation (ChIP) coupled with quantitative PCR (qPCR) was performed in U2OS cells with the pan anti-Kcr antibody to measure the Kcr levels near TSS of the candidate genes. We found that Sirt3 knockdown by siRNA resulted in significant increases in the Kcr levels of five of the seven genes analyzed, indicating that Sirt3 may directly regulate crotonylation dynamics at the genomic loci where it binds to (see Figure 5). Interestingly, the mRNA levels of the three candidate genes with increased Kcr levels, *Ptk2*, *Tshz3* and *Wapal*, were also significantly up-regulated upon Sirt3 knockdown (see Figure 5 ). Given that histone Kcr is enriched at active gene promoters and potential enhancers, this positive correlation between the gene transcription level and the nearby histone Kcr level upon Sirt3 knockdown suggests that Sirt3 might relieve a repressive effect on these target genes through ‘erasing’ histone Kcr ‘marks’. To examine this correlation genome-wide requires comprehensive profiling of global histone Kcr and gene expression regulated by Sirt3 using ChIP couple to high-throughput sequencing (ChIP-seq), in combination with RNA sequencing (RNA-seq) in future studies.

In addition, same type of PTM at different modification sites of histones may have distinct effects on gene expression. For example, trimethylation at histone H3 Lys-4 (H3K4Me3) ‘marks’ genes that are being actively transcribed, whereas the same modification at H3 Lys-27 (H3K27Me3) ‘marks’ transcriptionally silent chromatin. By analogy, it is possible that crotonylation at specific lysine sites of histones could also play different roles in the regulation of gene expression. The study of the effects of site-specific histone Kcr ‘marks’ (e.g., H3K4Cr) targeted by Sirt3 on the regulation of gene expression is an important next step. Unfortunately, in our preliminary ChIP-qPCR study, we found that the only commercial available antibody that target H3K4Cr (from PTM BioLabs) is not suitable for ChIP experiments (the immune-precipitated chromatin was less than 0.1% of input). Therefore, new antibodies for this purpose need to be developed in future studies.

*5) The structural studies of crotonylated peptide to SIRT3 propose a binding mode involving Phe180 that is highly conserved. What is not mentioned is that this residue is known to interact strongly with NAD in prior structural work (e.g. Wolbergers comprehensive studies, which include termolecular structures of both NAD and peptides bound to sirtuin active sites). The authors state that this residue is not conserved in SIRT4-7, but its identification and conservation has been supported by many sequence comparison and structural studies. The observed lack of decrotonylation activity upon mutation may therefore be a consequence of disrupted NAD binding rather than demonstrating substrate selectivity. It also questions whether the observed binding mode in the absence of NAD is relevant if NAD is bound to the enzyme. Steady state parameters for the decrotonylation activity of the mutant and binding studies of NAD and the crotonylated peptide to the mutant may be able to address these concerns*.

We thank the reviewers for this thoughtful comment. Indeed, Wolberger’s studies revealed that an invariant Phe33 in Sir2Tm (conserved in sirtuins and corresponding to human Sirt3 Phe157 rather than Phe180) strongly interacts with NAD. To the best of our knowledge, the interaction between Phe180 of Sirt3 and NAD have not been described. In our crystal structure of Sirt3-H3K4Cr complex, Phe180 is involved in recognition of crotonyl lysine via π−π stacking interaction. To further examine the importance of this conserved phenylalanine to the enzymes’ decrotonylase activity, we mutated Phe180 of Sirt3 to a leucine residue, which lacks an aromatic ring as a π-donor but remains a similar hydrophobicity. We then carried out kinetic studies on this F180L mutant Sirt3. The steady-state kinetics data showed that the catalytic efficiency of Sirt3 F180L mutant (*k*_cat_/*K*_m_ = 21 s^-1^ M^-1^) for the hydrolysis of the H3K4Cr peptide was about 40-fold lower than that of wild-type Sirt3 (*k*_cat_/*K*_m_ = 783 s^-1^ M^-1^) (see Figure 4—figure supplement 1), indicating a critical role of the phenylalanine-mediated π−π interaction in the enzymes’ decrotonylation activity. Interestingly, the F180L mutation caused only around 2-fold decrease in the enzyme’s deacetylation activity (see Figure 4—figure supplement 6). This result rules out the possibility that the observed significant decrease in the enzyme’s decrotonylase activity is caused by a potential disruption of the NAD-binding pocket in the mutated Sirt3.